# Advanced $NO_2$ retrieval technique for the Brewer spectrophotometer applied to the 20-year record in Rome, Italy

Henri Diémoz[1], Anna Maria Siani[2], Stefano Casadio[3], Anna Maria Iannarelli[3], Giuseppe Rocco Casale[2,a], Vladimir Savastiouk[4], Alexander Cede[5,6], Martin Tiefengraber[5,7], and Moritz Müller[5]

[1]Regional Environmental Protection Agency (ARPA) of the Aosta Valley, Saint-Christophe, Italy
[2]Physics Department, Sapienza University of Rome, Rome, Italy
[3]Serco Italia, Rome, Italy
[4]International Ozone Services Inc., Toronto, Ontario, Canada
[5]LuftBlick, Innsbruck, Austria
[6]NASA Goddard Space Flight Center, Greenbelt, USA
[7]Department of Atmospheric and Cryospheric Sciences, University of Innsbruck, Innsbruck, Austria
[a]now independent researcher

**Correspondence:** Henri Diémoz (h.diemoz@arpa.vda.it)

**Abstract.** A re-evaluated data set of nitrogen dioxide ($NO_2$) column densities over Rome for the years 1996 to 2017 is here presented. This long-term record is obtained from ground-based direct sun measurements with a MkIV Brewer spectrophotometer (serial number #067), further reprocessed using a novel algorithm. Compared to the original Brewer algorithm, the new method includes updated $NO_2$ absorption cross sections and Rayleigh scattering coefficients, and accounts for additional

atmospheric compounds and instrumental artefacts, such as the spectral transmittance of the filters, the alignment of the wavelength scale and internal temperature. Moreover, long-term changes in the Brewer radiometric sensitivity are tracked using statistical methods for in-field calibration. The resulting series presents only few (about 30) periods with missing data longer than one week and features $NO_2$ retrievals in more than 6100 days, covering nearly 80 % of the considered 20-year period. The high quality of the data is demonstrated by two independent comparisons. In a first intensive campaign, Brewer #067 is com-

pared against another Brewer (#066), recently calibrated at the Izaña Atmospheric Observatory through the Langley method and there compared to reference instrumentation from the Network for the Detection of Atmospheric Composition Change (NDACC). Data from this campaign show a highly significant Pearson's correlation coefficient of 0.90 between the two series of slant column densities, slope 0.98 and offset 0.05 DU ($1.3 \times 10^{15}$ molec cm$^{-2}$). The average bias between the vertical column densities is 0.03 DU ($8.1 \times 10^{14}$ molec cm$^{-2}$), well within the combined uncertainty of both instruments. Brewer #067 is

also independently compared with new-generation instrumentation, a co-located Pandora spectrometer (#117), over a 1-year long period (2016-2017) at Sapienza University of Rome, showing linear correlation indices above 0.96 between slant column densities, slope of 0.97 and offset of 0.02 DU ($5.4 \times 10^{14}$ molec cm$^{-2}$). The average bias between vertical column densities is negligible (-0.002 DU or $-5.4 \times 10^{13}$ molec cm$^{-2}$). This, incidentally, represents the first intercomparison of $NO_2$ retrievals between a MkIV Brewer and a Pandora instrument. Owing to its accuracy and length, the Brewer data set collected in Rome can

be useful for satellite cal/val exercises, comparison with photochemical models, and for better aerosol optical depth estimates ($NO_2$ optical depth climatology). In addition, it can be employed to identify long-term trends in $NO_2$ column densities over

a metropolitan environment, during two decades witnessing important changes in environmental policies, emission loads and composition, and the effect of a worldwide economic recession, to offer just a few examples. The method can be replicated on the more than 80 MkIV spectrophotometers operating worldwide in the frame of the international Brewer network. The $NO_2$ data set described in this manuscript can be freely accessed at https://doi.org/10.5281/zenodo.4715219 (Diémoz and Siani, 2021).

# 1 Introduction

Few trace gases are as ubiquitous in the atmospheric chemistry as nitrogen oxides. These are released by both natural sources, such as microbiological activity, biomass burning, lightning and volcanoes (Bates and Hays, 1967; Lee et al., 1997; Schumann and Huntrieser, 2007), and anthropogenic sources, like high-temperature combustion in vehicle engines, power plants, industry, and use of fertilisers. The different nitrogen oxides species are transported and mixed, not without reacting, into the different layers of the atmosphere. Among them, nitrogen dioxide ($NO_2$), one of their most abundant forms, is involved in the catalytic cycles accounting for almost half of the ozone removal by gas-phase reactions in the high stratosphere, between 25 and 40 km altitude (Crutzen, 1970; Garcia and Solomon, 1994), while below 25 km it moderates the ozone loss by converting active ozone-depleting species into inactive reservoir forms (Chartrand et al., 1999; Portmann et al., 1999). In the lower troposphere, nitrogen oxides control the abundance of ozone through the reactions leading to photochemical smog (Haagen-Smit, 1952), participate to aerosol (nitrates) formation (Seinfeld and Pandis, 2006), and can be adsorbed on dust particles (Zhou et al., 2015). Since they are also directly harmful to human health (Krzyzanowski and Cohen, 2008) and to the environment (Bell and Treshow, 2002), nitrogen oxides are notoriously known air pollutants and routinely monitored in the ambient air, to evaluate compliance with air quality standards, by environmental protection agencies. Last, but not least, owing to its ability of absorbing solar radiation in the ultraviolet (UV) and visible bands, $NO_2$ is also known for its radiative effect (Solomon et al., 1999) and for its capacity of interfering with the estimate of other atmospheric compounds in solar photometry (Shaw, 1976).

Based on its peculiar spectral absorption of solar radiation, the $NO_2$ vertical column density (VCD) can be estimated by ground-based spectroscopy (Brewer and McElroy, 1973; Noxon, 1975; Kerr et al., 1977) and from spaceborne instruments (Richter et al., 2005; Celarier et al., 2008; Griffin et al., 2019). Remote sensing techniques are able to probe the whole atmospheric column (or specific portions of it), whereas surface in-situ instrumentation (e.g., air quality monitors) provides information on the $NO_2$ burden at the surface and is sensitive to the mixing layer height (Petritoli et al., 2004). Estimates from satellite radiometers are usually associated with large uncertainties over complex terrains or very polluted areas (Boersma, 2004; Leitão et al., 2010) due to assumptions on the air mass factors based on climatologies, and might underestimate the local amounts owing to their generally low spatial and temporal resolution (Herman et al., 2019; Judd et al., 2019). For example, Verhoelst et al. (2021) found strong negative biases of $-50$ % in the $NO_2$ total columns measured by Copernicus Sentinel-5P TROPOMI over Rome compared to ground-based Pandora spectrometers. Hence, improvement of observations from space requires a continuous validation of satellite retrievals against column densities from ground-based, remote-sensing instrumentation.

However, whilst surface air quality monitors and satellites often benefit from long-term records, accurate multi-decennial data sets from ground-based spectrometers are not yet widely available (Wang et al., 2020; Gruzdev and Elokhov, 2021). Such series are also beneficial for assessing trends in the $NO_2$ stratospheric concentration (Gruzdev, 2009; Werner et al., 2013), e.g. for climate studies, and for evaluating the abatement effects on the tropospheric loads resulting from environmental policies and emission controls, vehicle fleet renovation, and even worldwide crises such as the recent (2008) economic recession or the confinement due to SARS-CoV-2 pandemic (Castellanos and Boersma, 2012; Russell et al., 2012; Hilboll et al., 2013; Vrekoussis et al., 2013; Bassani et al., 2021; Diémoz et al., 2021; Macdonald et al., 2021).

Among other instruments to measure $NO_2$ columns, such as the Système d'Analyse par Observation Zénithale (SAOZ) employed in the Network for the Detection of Atmospheric Composition Change (NDACC) and the Pandora spectrometers constituting the Pandonia Global Network (PNG), the Brewer global network represents a possible, and undervalued, source of information on total nitrogen dioxide column amounts. Indeed, $NO_2$ estimates from both direct sun and zenith sky observation geometries have been proven possible with Brewers spectrophotometers, beside ozone retrievals for which they are well-known reference instruments. Cede et al. (2006), for example, used MkIII Brewers to retrieve nitrogen dioxide VCDs from direct sun measurements in the UV-A range (nominally, 315-400 nm), although the spectral structure of the $NO_2$ absorption cross sections at these wavelengths is rather weak and does not provide high signal-to-noise ratios. MkIV Brewers, on the contrary, were conceived in 1985 for the exact purpose of deriving both ozone in the UV-B (nominally, 280-315 nm) and nitrogen dioxide in the visible range (wavelengths larger than 400 nm), which enhances the Brewer sensitivity to this trace gas owing to the pronounced spectral structures of the $NO_2$ cross section in that band (Kerr, 1989). We estimate that more than 50 MkIV Brewer spectrophotometers from 22 countries have been collecting raw (unprocessed) data in the visible range and sending them to the World Ozone and Ultraviolet radiation Data Center (WOUDC) since the establishment of the Brewer network in the 1990's. These data sets can be potentially used to extract information on past $NO_2$ total columns. Moreover, another 30 MkIV Brewers have been programmed to retrieve ozone VCDs only, and can start collecting $NO_2$ data from now on, which would enlarge the potential network to more than 80 spectrophotometers. Accurate $NO_2$ data processing from the Brewer, however, is hampered by the lack of algorithm updates in the operating software since the 1980's and by the absence of a travelling $NO_2$ reference such as for ozone. Nevertheless, considerable progress have been made in the last decades by Barton (2007) and Diémoz et al. (2014b), who improved the Brewer $NO_2$ algorithm and applied it to short-term series for demonstration purposes.

In this study, we prove that it is possible to extract accurate information on $NO_2$ VCDs even from long-term data sets collected with MkIV Brewers. To this end, we employ the Brewer spectrophotometer (serial number #067) operating at the Atmospheric Physics Laboratory (APL), located at the Physics Department of Sapienza University of Rome. The instrument has been providing UV global irradiance (Campanelli et al., 2019; Vitt et al., 2020) and ozone retrievals (Siani et al., 2018) since 1992 and has been measuring direct solar radiation in the visible range for $NO_2$ estimates since 1996 (Francesconi et al., 2004). This series is of particular interest owing to the characteristics of the station, located in the Italian capital city, a populated and trafficked metropolis, and to the continuity and length of the data set. We reprocess this series based on the algorithm developed by Diémoz et al. (2014b, this publication being hereafter referred to as "reference paper", or RP2014),

further improved to reduce the Brewer sensitivity to some instrumental parameters such as the alignment of the wavelength scale and the Brewer internal temperature. Radiometric sensitivity changes with time are also tracked using state-of-the-art methods for calibration in the field (since no other proxy or reference instrument is available during the whole period). The series analysed here covers the period 1996-2017, however measurements are still being performed and will be processed in a future study. An intensive intercomparison campaign with another MkIV Brewer recently calibrated at a pristine site through the Langley method and a 1-year long comparison with a more modern instrument, a collocated Pandora spectrometer, demonstrate the high quality of the Brewer data set in Rome.

The paper is structured as follows: Sect. 2 introduces the data and methods, i.e. the measuring site (Sect. 2.1), the involved instruments (Sect. 2.2) and the updates to the processing technique (Sect. 2.3). The new data set is then presented in Sect. 3.1, together with the description of the uncertainty estimation (Sect. 3.2). We compare the Brewer retrievals with other instruments in Sects. 3.3 and 3.4. Finally, the significance of the new data set and perspectives of further applications are discussed in Sect. 4.

The main results are expressed in both Dobson Units (DU), for direct comparison with previous literature on Brewers, and in $\mathrm{molec\,cm^{-2}}$, as common in the DOAS and satellite scientific communities. Furthermore, the final data set is also provided in $\mathrm{mol\,m^{-2}}$.

## 2   Data and methods

### 2.1   Site description

Brewer #067 has been operating since 1992 at Sapienza APL (41.901N, 12.516E, 75 m asl). The observatory is located on a wide terrace of the "Fermi" building on the university campus, in an urban context (Meloni et al., 2000). Rome, the Italian capital city, represents the most populated municipality in Italy, with almost 3 million inhabitants in the urban area, and more than 4 million inhabitants including the suburban areas. Nitrogen dioxide emissions are mainly imputable to human activities, such as road traffic, which in Rome is responsible for 80 % of the $NO_2$ emissions (Municipality of Rome, 2017) due to widespread use of private cars, and domestic heating. As a result, despite the $NO_2$ surface concentrations having nearly halved in the last 20 years (ARPA Lazio, 2020), exceedances of the annual limit value of $40\,\mathrm{\mu g\,m^{-3}}$ are still detected by some air quality stations in the city centre (ARPA Lazio, 2021). Other anthropogenic emission sources of nitrogen dioxide include air and sea traffic in proximity of the city, as well as combustion in energy and industry plant (Bassani et al., 2021). As for the natural sources, wildfire plumes might also occasionally contribute to the observed tropospheric $NO_2$ levels (Campanelli et al., 2021). The site is characterised by hot-summer Mediterranean climate (Csa, according to the Köppen-Geiger classification), also known as a "typical Mediterranean climate", with strong solar radiation expected to relevantly impact the $NO_2$ surface concentrations through photochemical processes, as also common in other southern European locations. Pollutant dispersion, and notably the near-surface and tropospheric amounts of nitrogen dioxide, are additionally influenced by temperature inversions, the heat island effect (Di Bernardino et al., 2021b) and by penetration of the breeze front in the urban area (Di Bernardino et al., 2021a).

In the last few years, APL serves as the main measuring headquarter of the Boundary layer Air Quality Using Network of INstruments (BAQUNIN) project (https://www.baqunin.eu/, last access: 26 April 2021), beside two additional stations in

semi-rural (CNR-ISAC, Tor Vergata) and rural (CNR-IIA, Montelibretti) context. Indeed, various ground-based instruments dedicated to monitoring of the main trace gases and aerosols along the atmospheric column and at the surface are in operation at APL, for purely scientific purposes or in the frame of calibration and validation of satellite products. Among them, a Pandora spectrometer (#117, Sect. 2.2.2) has been operating since 2016.

In June 2013, Brewer #067 was moved for one month to the atmospheric observatory of Aosta–Saint-Christophe, Italy (Diémoz et al., 2014a; Fountoulakis et al., 2020), in the northwestern Alps, for comparison with Brewer #066, routinely operated there. The observatory is located on the roof of the local Environment Protection Agency (45.742N, 7.357E, 570 m asl, WIGOS ID 0-380-5-1) at the bottom of an Alpine valley with a wide horizon, in a semi-rural context. The local pollution sources only include road traffic and wintertime domestic heating from the nearby city of Aosta (34,000 inhabitants). Although transport of $NO_2$ from the Po Valley, one European pollution hotspot, is observed to take place on fair-weather days owing to development of thermal winds (Diémoz et al., 2019), the tropospheric concentrations of this trace gas hardly ever reach large amounts (the yearly averages of $NO_2$ surface concentrations being about half of the annual limit value of $40\,\mu g\,m^{-3}$).

## 2.2 Instruments

The data set presented in this study is obtained from Brewer #067 in Rome, which is accurately described here below. In addition, we provide information about Brewer #066 and Pandora #117, whose $NO_2$ retrievals are compared to the ones from Brewer #067.

### 2.2.1 MkIV Brewer spectrophotometers

Brewer spectrophotometers, and notably MkIV Brewers, are described in detail elsewhere, e.g. in RP2014. In short, the instruments collect solar radiation directly from the sun ("direct sun mode") or from the zenith ("zenith sky mode"), depending on rotation of a reflective prism, which selects the elevation angle of the field of view. In the direct sun configuration, which is the only discussed in the present study, the sun light is directed on a diffusing filter and, possibly, on one of the available neutral-density filters, chosen so as to regulate the beam intensity to suitable levels for detection by a photomultiplier tube (PMT) in its linearity range. The filters available on a Brewer provide attenuations from 0 up to 2.5 decades, and are considered spectrally neutral in the default ozone and $NO_2$ retrieval algorithms. Actually, as discussed in RP2014, the filter transmittance is not spectrally flat, therefore corrections are needed to achieve high-accuracy retrievals, especially of weakly absorbing species such as $NO_2$. MkIV Brewers are "single" Brewers, i.e. equipped with only one monochromator, which disperses, through a holographic diffraction grating, the incoming radiation into its wavelength spectrum. A rotating slit mask allows the PMT to sequentially measure the solar spectral irradiance at up to six different wavelengths at high speed (0.1147 s integration time for each slit). Twenty slit cycles are scheduled for each individual measurement, and five individual measurements are then collected to obtain a direct sun retrieval (as the average of the samples) and its standard deviation, for a total of 100 cycles for each complete measurement loop. To retrieve the ozone total column, MkIV Brewers use the grating in the third order of diffraction, which provides output wavelengths in the ultraviolet range, and only solar irradiance recorded at the four longest wavelengths (about 310–320 nm) is employed in ozone retrievals. Conversely, to retrieve nitrogen dioxide, the diffraction

**Table 1.** Operating wavelengths and weightings ($\gamma_i$) employed with Brewer #067 for the retrieval of nitrogen dioxide. BNALG0 refers to the algorithm by Kerr (1989), BNALG1 to the algorithm by Diémoz et al. (2014b), and BNALG2 to the updates proposed in this paper.

| Wavelength (nm) | 425.02 | 431.40 | 437.35 | 442.83 | 448.08 | 453.20 |
|---|---|---|---|---|---|---|
| Weightings (BNALG0) | 0 | $1\times10^{-1}$ | $-5.9\times10^{-1}$ | $1.1\times10^{-1}$ | 1.2 | $-8.2\times10^{-1}$ |
| Weightings (BNALG1) | $4.353\times10^{-2}$ | $1.489\times10^{-1}$ | $-4.925\times10^{-1}$ | $-4.929\times10^{-2}$ | $7.534\times10^{-1}$ | $-4.041\times10^{-1}$ |
| Weightings (BNALG2) | $6.657\times10^{-2}$ | $2.632\times10^{-2}$ | $-2.528\times10^{-1}$ | $-2.603\times10^{-1}$ | $8.326\times10^{-1}$ | $-4.124\times10^{-1}$ |

grating is used in the second order, and the emerging visible light has wavelength ranging between 425 and 435 nm (Table 1), where the differential structure of the $NO_2$ absorption cross sections is pronounced.

Some tests are regularly alternated to measurements to check that the Brewer is stable over time. For example, due to temperature changes within the instrument, the wavelength alignment may vary during the day, hence tests using a line emission source (provided by an internal mercury lamp), representing a stable wavelength reference, are important to keep the wavelength registration within $\pm0.01$ nm. It is worth noting that, while the sensitivity of the Brewers to slight wavelength changes is minimised for ozone retrievals by choosing a suitable grating position ("sun scan test"), this is not possible for $NO_2$ retrievals because the solar and $NO_2$ absorption spectra do not give rise to a similar effect in the visible range. Along with the wavelength registration, the spectral sensitivity of a Brewer can also change with the age of an instrument. To track these changes and correct them, some "standard lamp tests" (SL test) are included in the daily measurement schedule. During the test, which can be carried out in both "ozone" and "nitrogen dioxide" modes, the irradiance from an internal halogen lamp is collected instead of the sun, and is processed similarly to the solar measurements. The results (called "F-ratios" for the $NO_2$ measuring mode, cf. Fig. 1 as an example) are then used to monitor the Brewer stability and correct the retrievals, if needed (Siani et al., 2018).

Two major events impacted on the sensitivity of Brewer #067. In July 2012, the instrument was temporarily moved to Arosa (CH) and took part in an international intercomparison. There, during the maintenance operations, focusing of the monochromator in the UV was improved by moving the mirror closer to the grating. This operation also likely affected focusing in the visible range and the energy distribution among the slits, thus introducing a strong discontinuity in the Brewer spectral sensitivity, though well compensated by the algorithm proposed here (Sect. 2.3.2). The second event occurred some years later, in July 2017, when extraordinary servicing was performed on the PMT, with the aim of improving the stability of ozone retrievals. Unfortunately, the procedure degraded the quality of $NO_2$ retrievals by introducing short-term instabilities, strongly dependent on the internal temperature in a non-linear way. Hence, we reprocess the data set only until that date to include retrievals with the highest possible accuracy in this data set. In the forthcoming future, the addition of a thick diffusing filter, which will lead to decreased photon counts, is expected to fix the issue.

Before carrying out $NO_2$ retrievals, both Brewers involved in this study, #066 (routinely operated in Aosta–Saint-Christophe) and #067, were accurately characterised. The wavelength dispersion function in the visible range is obtained from measure-

ments of emission lines in that band, as illustrated in RP2014. The stability of the dispersion function over the whole period is monitored using the UV dispersion, regularly measured during the audits, as a proxy. This test shows that the wavelength scale of Brewer #067 is stable within $\pm 0.03$ nm for the whole considered period (1996–2017). Although the new algorithm presented in this study (BNALG2, Sect. 2.3.1) is insensitive to such a small amount, the contribution to the overall uncertainty on $NO_2$ retrievals owing to slight wavelength changes is anyway taken into account in Sect. 3.2. The dispersion test is also employed to assess the spectral resolution provided by each slit and to degrade the absorption cross-sections to the instrumental resolution. The spectral transmittance of the density filters was assessed using the internal standard lamp in 2013. In principle, this measurement provides accurate estimates only for the thinnest filters, since the irradiance measured with the thickest filters is very low. However, the procedure was iterated every day for nearly two months, thus collecting a large number of samples. Moreover, to further check the accuracy of the filter characterisation, nearly-simultaneous retrievals using different filters are compared in the final data set, showing no systematic deviations. Nevertheless, we discuss the uncertainty on the filter transmittance in Sect. 3.2. In addition to the previous characterisation, Brewer #066 was furthermore calibrated for $NO_2$ retrievals at the Izaña observatory (Tenerife, Canary islands). This was achieved using a modified version of the Langley plot accounting for approximately daily linear trends in the stratospheric $NO_2$ VCD (due to photochemistry), as explained in RP2014.

The measurement schedule operating on Brewer #067 includes ozone measurements in direct sun and zenith sky geometries, Umkehr observations, UV scans, nitrogen dioxide measurements in both geometries, and tests. As a result, about 10–30 $NO_2$ measurements are performed each day, depending on the available solar zenith angles (SZAs) and sunshine hours.

### 2.2.2 Pandora spectrometer

The Pandora-2S instrument (serial number 117) employed in this study consists of two optical paths and spectrometers. The UV (VIS) spectrometer has a nominal wavelength range from 280 (380) to 530 (900) nm with an average spectral resolution of 0.6 (1.1) nm. For this study, only data from the UV channel are used. The field of view of the telescopes is 2.5° FWHM and the direct sun beam is further adjusted using a diffuser and, if needed, neutral density filters. The instrument has been characterized following the Pandonia Global Network (PGN) calibration procedures (Müller et al., 2020). The considered data correction steps (raw L0 data to corrected L1 data) are explained in the Blick Software Suite manual (Cede, 2019). Pandora direct sun $NO_2$ retrievals have been compared to spaceborne radiometer estimates and to other ground-based instruments during field campaigns (Herman et al., 2009; Wang et al., 2010; Flynn et al., 2014; Reed et al., 2015; Martins et al., 2016; Lamsal et al., 2017; Bösch et al., 2018; Tirpitz et al., 2021).

The Pandora data available for the present study, downloaded from http://data.pandonia-global-network.org (last access: 26 April 2021), are obtained using the direct sun retrieval code "nvs1" with Blick processor version 1.7 (Cede et al., 2021).

### 2.3 Nitrogen dioxide retrievals with Brewer #067

Accurate retrievals are generally the result of a reliable algorithm and careful calibration of the measuring instrument. Unfortunately, in the case of nitrogen dioxide, none of them is available for the vast majority of the MkIV spectrophotometers in the Brewer network. Indeed, the only algorithm implemented in the Brewer operating software is the one by Kerr (1989) (here-

after referred as to Brewer nitrogen dioxide algorithm version 0, or BNALG0 in short), based on now obsolete spectroscopic data sets, and relying on the characterisation of the specific spectrophotometer used by the authors (not necessarily similar to other instruments in the network). Previous studies found that retrievals using BNALG0 show both overestimations (McElroy et al., 1994; Barton, 2007) and random deviations (Hofmann et al., 1995), these latter due to interfering atmospheric species or instrumental artefacts, compared to different types of instruments. Moreover, no travelling standard calibrated for $NO_2$ re-

trievals is available to homogenise the retrievals within the network. In our reference publication, we focused on both issues, proposing an improved algorithm (hereafter referred as to BNALG1), specially developed for pristine sites, and a technique to calibrate MkIV Brewers at high-altitude, unpolluted stations. Here we aim at further overcoming the limitations of BNALG1, such as high sensitivity to wavelength misalignments, by introducing some improvements (BNALG2), and at testing statistical methods that sort out the issue of moving the Brewer to isolated locations in order to calibrate it, thus allowing the reprocessing

of historical data sets. A summary of the changes from BNALG0 to BNALG2 is provided in Table 2.

### 2.3.1 Algorithm

The $NO_2$ VCD ($X_{NO_2}$) is calculated from the fundamental Brewer equation, derived in Appendix A (Eq. A6) and reported here below (the term $\epsilon_U$ was dropped since we assume that it has been minimised):

$$X_{NO_2} = \frac{\sum_i \gamma_i \log I_{0_i} - \sum_i \gamma_i \log I_i^*}{\mu_{NO_2} \sum_i \gamma_i \sigma_{NO_2 i}} \qquad (1)$$

This is obtained from a linear combination of the Bouguer-Lambert-Beer law (Bouguer, 1729) at the various wavelengths $\lambda_i$ measured by the Brewer. The coefficients of the linear combination, $\gamma_i$, are chosen so that interference by known atmospheric absorbers is minimised (Eq. A2). $I_i^*$ represents the count rates, proportional to the direct sun irradiance, measured by the Brewer at the Earth surface. The asterisk indicates that this term can include slight adjustments for instrumental artefacts and interference by the atmospheric species (Eq. A4). $I_{0_i}$ represents the count rates that would be measured if no extinction took

place within the Earth atmosphere (Sect. 2.3.2). $\mu_{NO_2}$ is the $NO_2$ airmass factor (lengthening of the light optical path due to the inclination of the solar beam relative to the zenith, in the $NO_2$ layer) and $\sigma_{NO_2 i}$ is the $NO_2$ absorption cross section.

In BNALG0, the number of measured wavelengths employed in the algorithm is 5, which enables removal of interference by Rayleigh scattering, ozone, aerosols, and spectrally-flat factors. In BNALG1, we increased the used wavelengths to 6, i.e. all wavelengths routinely measured by the Brewer in the visible range. The spectroscopic data sets were also updated (Table 2),

and the respective cross sections are convoluted to the Brewer resolution taking the I0-effect into account (Aliwell et al., 2002). More absorbing species are considered (as adjustments in $I_i^*$), along with the non-neutrality of the density filters. The additional degree of freedom compared to BNALG0 is employed to maximise the Signal-to-Noise Ratio (SNR) by maximisation of the scalar product between the $NO_2$ cross section and the weightings (Eq. A7). One of the main unsolved issues of BNALG1 is the sensitivity to small wavelength misalignments, as discussed in the uncertainty estimation in RP2014. This is due to a deep

Fraunhofer line in the solar spectrum in the proximity of one of the measured wavelengths (431 nm) (Diémoz et al., 2016).

The algorithm proposed in the present article (BNALG2) includes the following improvements:

**Table 2.** Working principles of various algorithm for $NO_2$ direct sun retrievals with MkIV Brewers.

| | BNALG0 (Kerr, 1989) | BNALG1 (Diémoz et al., 2014b) | BNALG2 (present study) |
|---|---|---|---|
| Number of used wavelengths | 5 | 6 | 6 |
| $NO_2$ absorption cross sect. | Johnston and Graham (1976) | Vandaele et al. (2002), 220 K | Vandaele et al. (1998), 254.5 K |
| $O_3$ absorption cross sect. | Vigroux (1952) | Bogumil et al. (2003), 223 K | – |
| Rayleigh scattering coeff. | Brewer standard coeff. | Bodhaine et al. (1999) | Bodhaine et al. (1999) |
| Aerosols | Linear | $\lambda^{-1}$ | $\lambda^{-1}$ |
| Spectrally flat factors | Accounted for | Accounted for | Accounted for |
| $O_2-O_2$ absorption cross sect. | – | Hermans et al. (2003) | Hermans et al. (2003) |
| I0 effect (Aliwell et al., 2002) | – | Accounted for | Accounted for |
| SNR maximisation | – | Yes | Yes |
| Wavelength misalignments | – | – | Minimised |
| Filter non-neutrality | – | Accounted for | Accounted for |
| $NO_2$ effective layer height | 22 km | 27 km | 7.2 km |

1. To minimise the instabilities due to sensitivity to wavelength misalignments, we determine new weightings ($\gamma_i$) by including the spectral derivative of the solar spectrum in the fit (Eqs. A2–A3), as also done by Kerr (2002) and Cede et al. (2006). When the derivative of the solar spectrum is included in the Brewer fit, the absolute value of the weighting relative to the second wavelength ($\gamma_2$) correctly decreases (more than 5 times compared to BNALG1, i.e. from $1.489\times10^{-1}$ to $2.632\times10^{-2}$, Table 1), thus reducing, in the linear combination, the relative importance of solar irradiance collected at 431 nm wavelength. An example of the improved stability provided by BNALG2 compared to BNALG1 is given in Sect. 3.4 (Fig. 11) based on comparison with the Pandora spectrometer;

2. In order to keep one degree of freedom available for SNR optimisation and not to over constrain the system, we drop the ozone absorption from the fit (Eq. A2). We estimate that this triggers an error of about 0.01 DU ($2.7\times10^{14}$ molec cm$^{-2}$) on $NO_2$ for a 200 DU range of total ozone column amounts. Such a low sensitivity is due to the fact that the ozone cross sections are quite flat in the considered spectral range. If needed, that amount can be taken into account and corrected by considering the closest ozone retrieval in time, or even the daily average $O_3$, from the Brewer (in the UV range);

3. To make the retrievals by Brewer #067 consistent with the co-located Pandora spectrometer (Cede et al., 2021), the same spectroscopic data set as in the Pandora is employed (Table 2, rightmost column). The new cross sections at higher temperatures and the lower $NO_2$ effective height used for airmass calculations are more suitable, compared to BNALG1, to operation at polluted sites, where the contribution of tropospheric nitrogen dioxide can be relevant;

4. We found that Brewer $NO_2$ retrievals are slightly sensitive to the instrument internal temperature, similarly to ozone retrievals. Temperature sensitivity is not accounted for in either BNALG0 or BNALG1, and absolute temperature de-

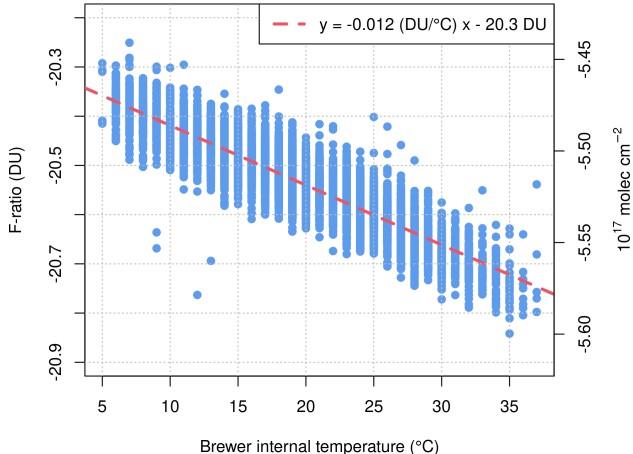

**Figure 1.** Results from the standard lamp (SL) test (blue dots, "F-ratio"), in both Dobson Units (DU, left axis) and $\mathrm{molec\,cm^{-2}}$ (right axis), as a function of the Brewer internal temperature. The slope of the regression line is used in BNALG2 to correct the Brewer sensitivity to the internal temperature in solar measurements. Only variations (differences) of the SL results on the y-axis, and not their absolute values, have a physical meaning.

pendence at only single wavelengths have been reported until now (e.g., Diémoz et al., 2016). Slight differences of the absolute sensitivity among wavelengths give rise to spectral dependence able to affect $NO_2$ retrievals. By analysing the multi-year data set of SL tests (Sect. 2.2.1) we are able to quantify this effect for Brewer #067 to $-0.012\,\mathrm{DU\,K^{-1}}$ ($-3.2\times10^{14}\,\mathrm{molec\,cm^{-2}\,K^{-1}}$, Fig. 1). We then use this coefficient to correct the linear combination of the count rates in solar measurements. Since the results of the SL tests after the temperature correction show very stable behaviour,

and since the extraterrestrial coefficients are retrieved on a regular basis (Sect. 2.3.2), we do not correct the calibration coefficient based on the results of the SL tests (as normally done for ozone measurements between one calibration and the next);

5. As in BNALG1, the interference by the oxygen dimer (O2−O2, with only a weak absorption band within the Brewer spectral range, at about 446 nm) is considered as a correction term to the retrieved $NO_2$ VCD, following Eq. (A4). This

is calculated from the Hermans et al. (2003) cross sections and the mean average pressure of the Rome site, and accounts for about 0.04 DU ($1\times10^{15}\,\mathrm{molec\,cm^{-2}}$).

### 2.3.2    Calibration

As shown by Eq. (1) and explained in Sect. 2.3.1, an extraterrestrial calibration (ETC) coefficient is needed to accurately retrieve the $NO_2$ column from spectrophotometric measurements. Furthermore, since the spectral sensitivity of a specific

instrument can change with time, this term cannot be determined once and for all, but a series of ETCs covering the whole length of the $NO_2$ data set is needed. The usual method employed for this purpose in well-established measurement networks

is to regularly transfer the calibration from a reference instrument (e.g., a travelling standard) to a field instrument. The former, in turn, is generally calibrated using the Langley plot technique (Langley, 1903) at a pristine, high-altitude site. Neither of the two methods can be used in the present case. First, a travelling Brewer acting as a reference for $NO_2$ calibrations does not exist. Second, Brewer #067 was never moved to a suitable site for a Langley campaign, and, even if this were the case, regular calibrations should have been organised throughout the 20-year long period to accurately track possible sensitivity changes with the instrument. As a consequence, to reprocess this long-term record, we are forced to use "field" calibration techniques based on statistical analysis of the data collected at Sapienza APL.

Herman et al. (2009) developed two methods for DOAS measurements that can be adapted to our case. Both methods rely on the fact that the total $NO_2$ column can be partitioned into a background contribution ($X_{NO_2}^b$, e.g. the stratospheric VCD), always present, and a variable fraction due to tropospheric pollution ($X_{NO_2}^p$). As the sum of $X_{NO_2}^b$ and $X_{NO_2}^p$ provides the total column, i.e. $X_{NO_2}$, Eq. (1) can be rewritten as

$$X_{NO_2}^b + X_{NO_2}^p = \frac{\sum_i \gamma_i \log I_{0_i} - \sum_i \gamma_i \log I_i^*}{\mu_{NO_2} \sum_i \gamma_i \sigma_{NO_2 i}} \tag{2}$$

By defining the measurement term, $F$, as

$$F \quad = \quad \frac{\sum_i \gamma_i \log I_i^*}{\sum_i \gamma_i \sigma_{NO_2 i}} \tag{3}$$

and the (hitherto unknown) extraterrestrial calibration (ETC) term as

$$ETC \quad = \quad \frac{\sum_i \gamma_i \log I_{0_i}}{\sum_i \gamma_i \sigma_{NO_2 i}}, \tag{4}$$

we can rearrange Eq. (2) in the following way:

$$F \quad = \quad ETC - \mu_{NO_2} \left( X_{NO_2}^b + X_{NO_2}^p \right) \tag{5}$$

Since we have divided the two terms in Eqs. (3)–(4) by the $NO_2$ differential absorption coefficient ($\sum_i \gamma_i \sigma_{NO_2 i}$), both $F$ and $ETC$ are now expressed in the same units as $X_{NO_2}$, e.g. DU or molec cm$^{-2}$.

The first method by Herman et al. (2009), called "Minimum-Amount Langley-Extrapolation" (MLE), assumes that the (a-priori unknown) background $NO_2$ VCD ($X_{NO_2}^b$) measurable along the atmospheric column is constant over a considered portion of the data set and that this minimum amount does not depend on the considered air mass. Hence, the upper envelope (lower envelope, in Herman's DOAS formulation) of $F$ against $\mu_{NO_2}$, i.e. (refer to Eq. 5)

$$F^b \quad = \quad ETC - \mu_{NO_2} X_{NO_2}^b \tag{6}$$

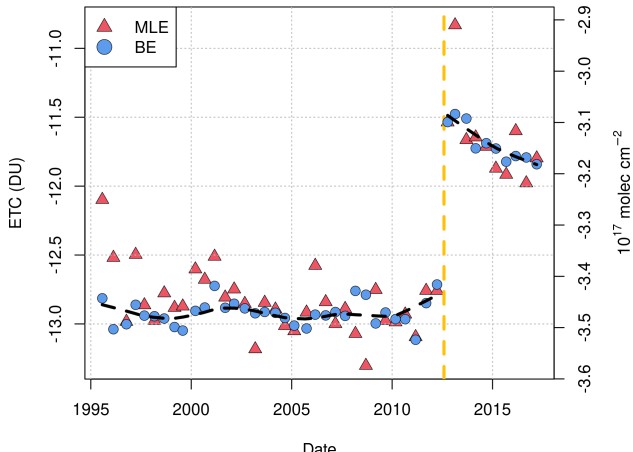

**Figure 2.** Extraterrestrial coefficients of Brewer #067 determined with the Minimum-Amoung Langley-Extrapolation (MLE, triangles) and Bootstrap-Estimation (BE, circles). The yellow vertical line indicates the discontinuity occurring in correspondence of the Arosa campaign in July 2012 (Sect. 2.2.1). The black dashed line represents a LOESS interpolation through the ETCs estimated by the BE method.

is well defined and corresponds to a straight line whose intercept is $ETC$. Conversely, the points below that envelope, i.e.

$$F \quad = \quad F^b - \mu_{NO_2} X^p_{NO_2} \tag{7}$$

are due to the additional, and variable, contribution of pollution ($\mu_{NO2} X^p_{NO_2}$) to the NO$_2$ slant column density.

In our implementation of the MLE, we group the available data into air mass bins (with at least 500 points per bin) and, to exclude outliers in the identification of the upper envelope, we determine the 97 % percentile of $F$ for each bin for the regression (corresponding to the third percentile in Herman's formulation). The regression over these points is performed employing a robust method, i.e. the iterated re-weighted least squares technique, implemented in the `rlm` function from the MASS library of the R package (Venables and Ripley, 2013). In order to track possible changes of the Brewer radiometric sensitivity with 315   time, we apply the MLE every six months (between the two solstices, to have sub-periods of similar air mass range), with a forced discontinuity in July 2012, when we clearly notice a sudden change (Sect. 2.2.1). Results from the MLE methods are shown in Fig. 2 with triangular markers. The average of the background VCD ($X^b_{NO_2}$), provided by the slope of the MLE regression (not shown), is 0.2±0.1 DU (5.4±2.7 ×10$^{15}$ molec cm$^{-2}$), which is similar to the value obtained by Herman et al. (2009) at Goddard Space Flight Center. This approximately corresponds to the expected average stratospheric NO$_2$ VCD. A 320   major drawback of this technique, however, is the large scatter of the ETCs with no apparent link to instrumental changes, which cannot be reduced even increasing the length of the sub-periods considered for the regression (e.g., up to several years).

     We therefore test the second statistical field method, the "Bootstrap Estimation" (BE) technique. In this case, the background NO$_2$ column is fixed on an a priori basis, and the corresponding slant column is added to $F$. By simply rearranging Eq. (5), we obtain

$$325 \quad F + \mu_{NO_2} X_{NO_2}^b \quad = \quad ETC - \mu_{NO2} X_{NO_2}^p \qquad (8)$$

We observe that the term at the right-hand side increases in pristine conditions reaching the maximum value of $ETC$ in the limiting case when $X_{NO_2}^p = 0$. Hence, $ETC$ can be simply calculated by a robust estimation of the maximum (e.g., 97 % percentile) of the quantity at the left-hand side of the equation above. We assume a background VCD $X_{NO_2}^b$ of 0.2 DU ($5.4 \times 10^{15}$ molec cm$^{-2}$), i.e. the average of the values previously determined by the MLE method. As apparent from Fig. 2, the BE method is more stable. Therefore, to obtain the final series of ETCs, we take the results of this last method, further interpolated with a locally estimated scatterplot smoothing (LOESS) curve to only keep the slowly-varying component of the calibration data set and not to introduce the short-term statistical noise (indeed, we have verified that these fast variations are not related to maintenance operations or changes in the instrument configuration).

It is worth noticing that the sudden variation in July 2012 (vertical dashed line in Fig. 2) is detected by both methods. If only an average ETC had been considered for the overall period, a step change larger than 1 DU or $27 \times 10^{15}$ molec cm$^{-2}$ (at $\mu_{NO_2} = 1$) in the NO$_2$ VCDs would have been introduced in the series. Instead, since we take that variation into account, no discontinuities can be noticed in the final NO$_2$ data set (Fig. 3).

### 2.3.3 Quality filter and cloudscreening

To ensure the highest retrieval accuracy and remove obviously erroneous measurements, always possible with automated instrumentation, we apply several checks:

1. A first set of controls is inspired by the Brewer Processing Software, developed by Environment and Climate Change Canada and described, e.g., by Siani et al. (2018). Raw counts lower than 2500 at the brightest slit, as well as differences between bright and dark counts lower than 250 (or 10 times the dark counts), are suggestive of incorrect pointing or very thick clouds. Measurements corresponding to these conditions are removed from the data set;

2. Spikes are corrected on the same principle as the Brewer operating software, i.e. by limiting the range of the measured count rates between 2 and $10^7$ photons s$^{-1}$;

3. To remove fast moving clouds between the sun and the observer, or erroneous measurements, the ratio between the standard deviation of the NO$_2$ retrieved from 5 consecutive samples and their average should be lower than 0.3. This empirical threshold is taken from previous studies (RP2014);

4. In the Brewers used here, the lower border of the quartz window casts a shadow on the zenith prism at solar zenith angles larger than 78 °. Thus, only direct sun measurements at SZA lower than this limit are analysed here.

The procedure described above does not filter out all clouds, and slowly moving, moderately thick clouds can still contaminate the data set. Therefore, we tighten the threshold for the first criterion and we consider only measurements characterised

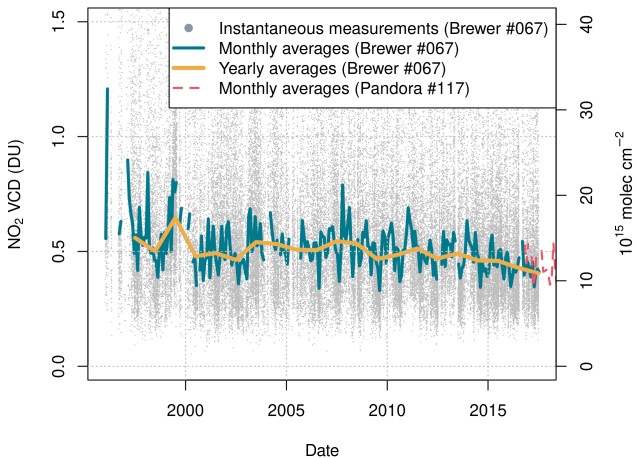

**Figure 3.** VCDs retrieved using BNALG2 from Brewer #067: instantaneous measurements (grey dots), monthly and yearly averages (continuous lines). The monthly averages of the retrievals from Pandora #117 (Sect. 3.4), operating at APL since 2016, are also shown for comparison (dashed line).

by count rates at the brightest slit, compensated for the filter transmittance, higher than $10^6$ photons $s^{-1}$. This threshold is
determined by examination of the frequency histogram of the count rates, which shows two distinct distributions, likely corresponding to clear-sky and residual clouds, well separated around the chosen threshold. Detailed visual inspection of the data set, together with comparison with Pandora (Sect. 3.4), confirm the accuracy of the quality screening algorithm.

## 3    Presentation of the re-evaluated data set

### 3.1    NO$_2$ vertical column densities

The NO$_2$ retrievals from Brewer #067 derived from algorithm BNALG2 are shown in Fig. 3. The record presents only few periods (about 30) with missing data longer than one week, and mostly concentrated in the initial period of NO$_2$ measurements. It includes more than 6100 days, covering nearly 80 % of the considered 20-year period. 95 % of the instantaneous VCDs (single points in the figure) range from 0.2 to 1.6 DU (5.4 to 43 $\times 10^{15}$ molec cm$^{-2}$). The baseline VCDs are rather constant throughout the observation period, as expected from the application of the BE technique, and their value reflect the amount
given as input to the calibration method. Measurements accumulate in summer, thanks to the wider daily range of air masses and – likely – to the sunnier weather and become sparser in winter. Monthly (yearly) averages, calculated when at least 50 % of the daily (monthly) averages are available, are also shown on the same figure as lines. A decreasing trend, still to be quantitatively assessed and discussed in more depth in future studies, is clearly perceptible, as expected from the tropospheric NO$_x$ emission reductions.

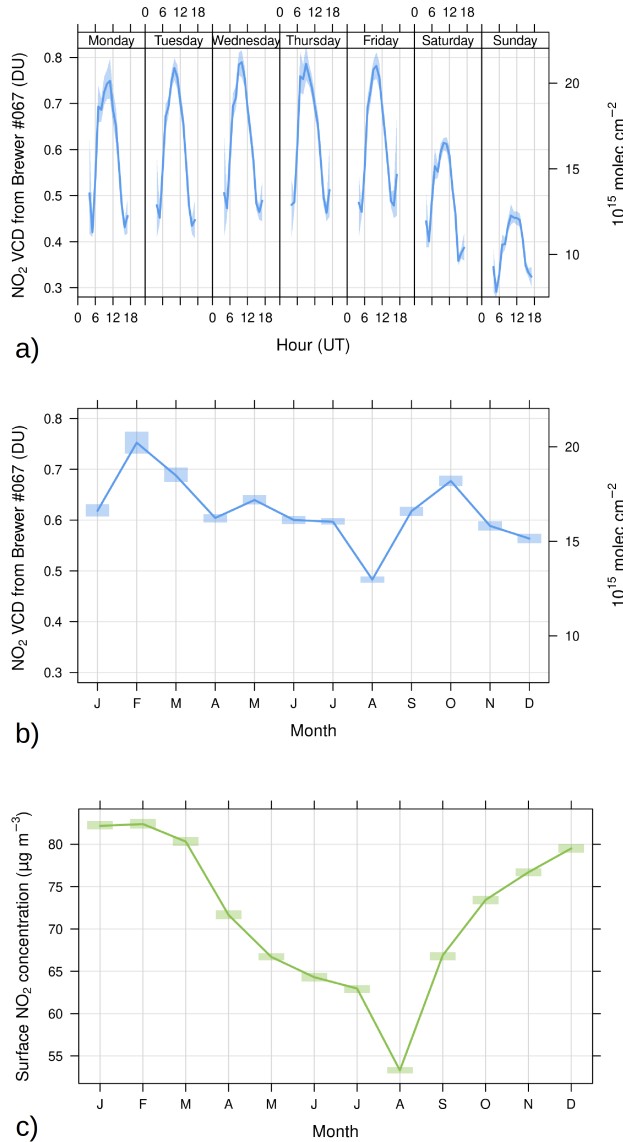

**Figure 4.** Daily, weekly (a) and monthly (b) cycles of $NO_2$ VCDs from Brewer #067 (averages from period 1996-2017), and monthly cycle of $NO_2$ surface concentrations (averages from period 1999-2017) measured by the local Environment Protection Agency at the Magna Grecia station (c), 2.6 km away from APL.

The daily and weekly average cycles of Brewer retrievals are represented in Fig. 4a, to demonstrate the reasonableness of the retrieval results. The most apparent feature is the inverse-U daily shape in the VCD amounts. This behaviour is not an indication of a wrong ETC, since the cycle is not centred at the solar noon (minimum air mass factor). Rather, as also detected by surface air quality monitors (not shown), the maximum is defined by the coupled effect of increased $NO_x$ emissions at

the surface during the morning rush hours and photochemical reactions involving generation of $NO_2$. Moreover, the peak is shifted towards midday on Sundays, as expected from a later start of the activities on such a day of the week. The beginning of another ascending phase is visible in the late afternoon of each day, however the effect of the measurement schedule and of the sunset does not enable the Brewer to sample the whole secondary peak. The same figure also illustrates the weekly cycle, with higher loads during work days and minima during the week end. Figure 4 additionally shows the monthly evolution of the $NO_2$ VCDs (b), as an average of the whole 1996-2017 period, and of the surface concentrations (c) determined by the local Environment Protection Agency in the period 1999-2017 at the Magna Grecia station, 2.6 km away from APL. The two plots share common features, the most evident being a minimum in August, the typical summer holiday month. However, the evolution of column densities and surface concentrations remarkably differs in winter. This is an expected feature and depends on the fact that spectrophotometric estimates are representative of the whole atmosphere while surface concentrations are sensitive to the mixing layer height, and thus to more stable meteorological conditions and reduced tropospheric mixing in winter. Moreover, the sampling time differs between the two techniques, since Brewer retrievals are only available during daylight hours, whilst surface data are collected all day round, this difference becoming maximum in winter.

A more in-depth analysis of the re-evaluated data is out of the scope of this paper and will be the subject of future investigations.

## 3.2 Uncertainty estimation

The uncertainty estimation of $NO_2$ retrievals from the Brewer is complex owing to the numerous factors contributing to the uncertainty in both the calibration and measurement phases, and to their respective correlations. For this reason, in RP2014 we took advantage of a Montecarlo technique to characterise the uncertainty of a Brewer calibrated with the Langley method at a pristine site, and then used at the same site or in a more polluted environment. In the present study, however, both the calibration procedure and the purpose of the uncertainty estimation are different. In particular, here we are not interested in a typical and general uncertainty assessment, but we want every measurement to be accompanied by its respective uncertainty, which is needed, for example, if the data set is employed for data reanalysis or trend assessment purposes. Hence, we use analytical formulas (rather than a Montecarlo technique) and we consider the calibration and the measurement uncertainties separately, since in this case the former is primarily driven by statistical fluctuations and a-priori assumptions.

1. Calibration. The ETC values from the BE calculation depend on some assumptions, such as the chosen percentile (e.g., selecting the 99th or the 95th percentile instead of the 97th already introduces differences of about $\pm 0.05$ DU or $1.3 \times 10^{15}$ molec cm$^{-2}$) and the background VCD $X_{NO_2}^b$. Changes of $\pm 0.05$ DU in this latter translate into differences of $\pm 0.1$ DU ($2.7 \times 10^{15}$ molec cm$^{-2}$) on the ETC values. The choice of the air mass range also influences the results for another 0.05 DU due to differences in the sampled set and, possibly, to systematic diurnal changes of the $NO_2$ tropospheric (and stratospheric) column, already mentioned above. Given the fact that these boundaries represent extreme conditions (and can be thus assimilated to the extremes of a uniform distribution), and that the above factors are probably correlated, we think that a (1 standard deviation) value of 0.08 DU ($2.2 \times 10^{15}$ molec cm$^{-2}$) is a conserva-

tive estimate of the overall ETC uncertainty. This value also reflects the experimental ETC variability in periods when the instrument is believed to be stable. This estimate, however, is higher than the (2-sigma) value of 0.05 DU assumed by Herman et al. (2009) for similar field calibration methods. This translates into an uncertainty of $0.08$ $\mathrm{DU} \cdot \mu_{NO_2}^{-1}$ on VCDs;

2. Radiometric stability. Since the ETCs are determined on a regular basis, and since results from the SL tests, after correcting for temperature, are very stable within the considered sub-periods, we do not consider any further uncertainty resulting from Brewer instabilities over time;

3. Poisson noise. The effect of instrumental noise on $NO_2$ VCDs is estimated using Poisson statistics on the recorded count rates (accumulated from 20 cycles for each sample) and propagating the resulting uncertainty to the linear combination of the logarithm of the count rates. Both bright measurements and dark count measurements are considered, and the results are further divided by $\sqrt{5}$, since each measurement loop is repeated 5 times. The contribution of Poisson noise on $NO_2$ VCDs (about 0.03 DU or $8.1 \times 10^{14}$ $\mathrm{molec\,cm}^{-2}$ at air mass factor close to 1 and less than 0.01 DU or $2.7 \times 10^{14}$ $\mathrm{molec\,cm}^{-2}$ at the largest SZAs, this behaviour originating from the contrasting effect of stronger light at low SZAs and increased $NO_2$ absorption at larger air masses) is lower than the one stated in RP2014. Indeed, in that preliminary study the number of data averaged in each measurement loop was reduced from 100 to 16 for practical purposes, i.e. to obtain more Langley measurements at twilight, when the air mass factor changes faster. If the values obtained in the present work are multiplied by a factor $\sqrt{100}/\sqrt{16} = 2.5$, then the same results as in RP2014 are found. Comparison of the expected Poisson uncertainty with the experimental standard deviation of the mean of the 5 samples provides similar results, with discrepancies smaller than 0.004 DU ($1.1 \times 10^{14}$ $\mathrm{molec\,cm}^{-2}$) presumably due to additional atmospheric variability during the measurements (not taken into account by Poisson statistics on count rates), thus confirming that the estimate of the instrumental noise is realistic;

4. Differential absorption coefficient. As for the uncertainty related to $NO_2$ cross sections, Vandaele et al. (2002) report values of 4–5 %. However, most of the factors influencing that estimation are likely to affect all wavelengths in a correlated way, therefore the propagation of that uncertainty on the differential absorption coefficient, as used in the Brewer algorithm, is lower. In our case, the main sources driving the uncertainty of the differential absorption coefficient are the unknown effective temperature of the $NO_2$ layer and the characterisation of the instrumental resolution used to scale the $NO_2$ cross sections. Concerning the first factor, an extreme variability range of $\pm 25$ K around the 254.5 K temperature used for the evaluation of the cross sections would lead to a 1-sigma uncertainty of 4 % on the absorption coefficient. This estimate is obtained in the case of a uniform temperature probability distribution and assuming a temperature dependence of the absorption coefficient of 0.285 % $\mathrm{K}^{-1}$, as can be derived from the Vandaele et al. (2002) spectroscopic data set using the weighting coefficients from Table 1. As for the second factor, a similar amount should be added if an uncertainty of 10 % is assumed on the characterisation of the Brewer resolution, as in RP2014. Accounting for both factors translates into an uncertainty of $0.06 \cdot X_{NO_2}$ on the retrieved column;

5. Neutral density filters. To check whether the spectral transmission of every neutral density filter is correctly characterised, all $NO_2$ retrievals from consecutive measurements (within an interval of 30 minutes maximum) using different filters are compared and their average differences are studied to identify any systematic offset. The average discrepancy between nearly-simultaneous data collected with different filters is smaller than 0.02 DU ($5.4 \times 10^{14}$ molec cm$^{-2}$). This will be taken as the uncertainty on the neutral density filters, thus accounting for 0.02 DU$\cdot\mu_{NO_2}^{-1}$ on VCDs;

6. Wavelength misalignments. This is one of the largest sources of uncertainty in the BNALG1 algorithm, accounting for about 0.1 DU ($2.7 \times 10^{15}$ molec cm$^{-2}$) at low SZAs for an uncertainty of $\pm 2$ steps in the micrometer operating position (RP2014). Conversely, BNALG2 was already conceived to minimise this factor according to Eq. (A3). To verify if the algorithm updates are successful, we perform radiative transfer simulations using the new weightings (Table 1), and we find that the effect of wavelength misalignments is reduced by about 20 times compared to BNALG1. For example, when the wavelength scale is shifted by $\pm 2$ microsteps, a maximum discrepancy of only 0.006 DU ($1.6 \times 10^{14}$ molec cm$^{-2}$) is found (at low solar zenith angles), while for an extreme change of $\pm 4$ microsteps the effect amounts to 0.018 DU ($4.8 \times 10^{14}$ molec cm$^{-2}$). The contribution of this factor to the overall uncertainty is then set to 0.01 DU$\cdot\mu_{NO_2}^{-1}$ ($2.7 \times 10^{14}$ molec cm$^{-2}\cdot\mu_{NO_2}^{-1}$), and represents one major improvement compared to BNALG1;

7. Oxygen dimer. $NO_2$ retrievals with BNALG2 are corrected for interference by O2$-$O2. Assuming a 30 % uncertainty on the oxygen dimer cross sections, as done in RP2014, the respective contribution to the $NO_2$ uncertainty is of the order of 0.01 DU ($2.7 \times 10^{14}$ molec cm$^{-2}$). This value is added to the overall uncertainty;

8. Unaccounted absorbers. Ozone and water vapour are not explicitly included in the fit, although they can interfere with the retrieval. According to Eq. (A7), a change of $\pm 200$ DU in ozone triggers an error of about 0.01 DU ($2.7 \times 10^{14}$ molec cm$^{-2}$) in NO2. The maximum estimate of the interference effect for water vapour is obtained by calculating the water vapour optical depth under saturation conditions at all altitude levels. This is performed using the HITRAN 2008 database (Rothman et al., 2009). Under these circumstances, the current algorithm might underestimate the $NO_2$ VCD by 0.08 DU ($2.2 \times 10^{15}$ molec cm$^{-2}$). However, it must be noted that the final data set is cloudscreened and that water vapour saturation conditions rarely occur in Rome in sunny conditions. Thus, a value of 0.02 DU ($5.4 \times 10^{14}$ molec cm$^{-2}$) accounting for both ozone and water vapour is added to the combined uncertainty only as a precautionary measure;

9. Air mass factor. The $NO_2$ air mass factor is a fundamental parameter in the retrieval (Eq. 1). The data analysed here only extend to solar zenith angles lower than 78 $^\circ$, as explained in Sect. 2.3.3. At these large angles, the air mass factor might be sensitive to the $NO_2$ effective height. If an uncertainty of $\pm 5$ km (as in RP2014) is taken as representative for the effective layer height, this leads to a maximum uncertainty of about $0.015 \cdot X_{NO_2}$ on VCDs at very large SZAs. Moreover, Since the Brewer PC is synchronised to the universal time by Network Time Protocol, we assume no uncertainty on the registration time and negligible uncertainty on the calculation of the SZA;

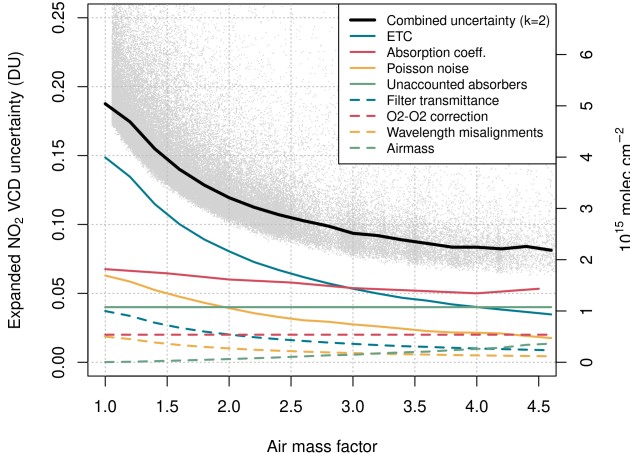

**Figure 5.** Total expanded uncertainty (black line) and its components, discussed in the text. The lines represent the median values for each air mass bin. Uncertainties are actually estimated for each individual measurement, here represented as grey dots.

10. Instrumental temperature. Analysis of SL tests in different periods leads to differences in the determination of the temperature coefficient lower than $\pm 5 \times 10^{-4}$ DU K$^{-1}$ ($1.3 \times 10^{13}$ molec cm$^{-2}$ K$^{-1}$). Even for temperature extremes of $\pm 20$ K from the average, the resulting uncertainty is lower than 0.01 DU ($2.7 \times 10^{14}$ molec cm$^{-2}$), and thus negligible. Moreover, visual inspection of temperature measurements in the instrument does not reveal any trend in more than 20 years, therefore we assume that the response by the internal temperature sensor is stable;

11. Based on RP2014, the uncertainties related to the photomultiplier dead time, to the assumed aerosol Ångström exponent, to spectral and geometrical straylight, and to the Rayleigh correction in the visible range are deemed negligible.

Figure 5 depicts the expanded ($k = 2$) measurement uncertainty and its components previously described as a function of the air mass factor, which is the dominant factor of variability for VCD uncertainties. For sake of clarity, the uncertainties associated to each NO$_2$ retrieval (grey dots) are grouped in air mass bins and their median is plotted as a line. The expanded uncertainties of individual measurements mostly rely between 0.06 DU ($1.6 \times 10^{15}$ molec cm$^{-2}$) at large SZAs and 0.23 DU ($6.2 \times 10^{15}$ molec cm$^{-2}$) at small SZAs, and are much lower (less than 50 %) than the ones obtained with BNALG1 in RP2014.

### 3.3 Comparison with Brewer #066

To illustrate the accuracy of the new series, we compare two subsets of the BNALG2 record with simultaneous NO$_2$ estimates obtained with reference instrumentation. A first campaign is organised from 3 to 27 June 2013, when Brewer #067 was moved to the Aosta–Saint-Christophe observatory and compared to Brewer #066. This latter, in turn, was calibrated in September and October 2012 at the Izaña observatory (Canary island), where its retrievals were also validated against two instruments from the NDACC network: a Fourier Transform Infrared Spectrometer and a RASAS-II Multi Axis Differential Optical Absorption

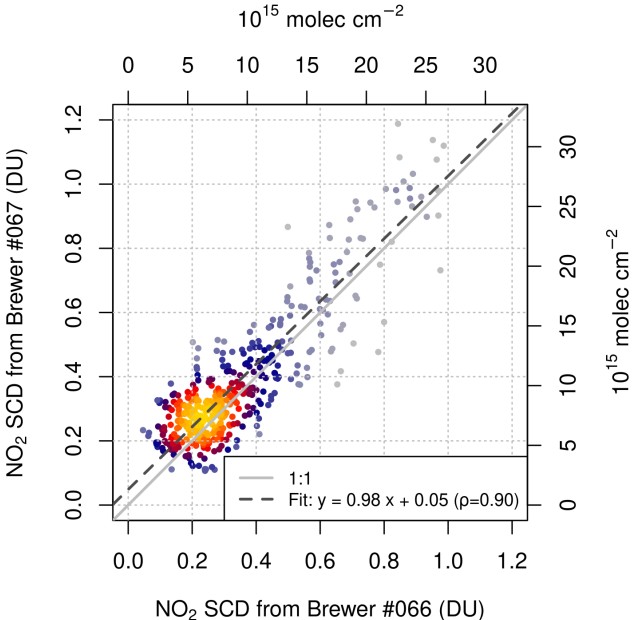

**Figure 6.** Comparison of the $NO_2$ slant column densities retrieved from Brewer #067 and Brewer #066 (considered as the reference) at the Aosta–Saint-Christophe observatory. Colours represent the density of the points based on a two-dimensional Kernel Density Estimation (Venables and Ripley, 2013). The Pearson's correlation coefficient between the two SCD series is $\rho = 0.90$.

Spectroscopy (MAX-DOAS) spectrometer (cf. RP2014 for further details). In the months following the Langley calibration, the stability of Brewer #066 was carefully checked through the internal (SL) tests.

Brewers #066 and #067 were programmed to take simultaneous measurements in both direct sun and zenith sky geometries. Here we only consider direct sun data collected on fair weather days, which consist in 445 simultaneous retrievals. Since measurements from Brewer #066 were validated in Izaña using BNALG1, we do not reprocess them with BNALG2, but we

only rescale them to the new differential absorption coefficient and use the new $NO_2$ effective layer height in the air mass factor formulation (Table 2). The retrievals from Brewer #067 are instead performed with BNALG2, as in Rome.

The maximum $NO_2$ Slant Column Density (SCD), i.e. the product between VCD and air mass (which is the real quantity the two instruments are sensitive to), estimated by both Brewers in Aosta–Saint-Christophe is 1.3 DU ($35 \times 10^{15}$ molec cm$^{-2}$). This first comparison features a Pearson's correlation coefficient of 0.90 between the two sets (Fig. 6, p-value $< 2.2 \times 10^{-16}$, slope

0.98 and offset 0.05 DU, or $1.3 \times 10^{15}$ molec cm$^{-2}$). VCDs derived with Brewer #067 are slightly larger than the ones from Brewer #066 (by 0.03 DU or $8.1 \times 10^{14}$ molec cm$^{-2}$), however it must be considered that Poisson's noise already accounts for 0.03 DU (at $\mu_{NO_2} = 1$) and that the uncertainty related to wavelength misalignments with BNALG1 (Brewer #066) accounts for 0.08 DU ($2.2 \times 10^{15}$ molec cm$^{-2}$). Thus, the mean bias is well within the combined uncertainty, and the comparison results can be defined very satisfactory.

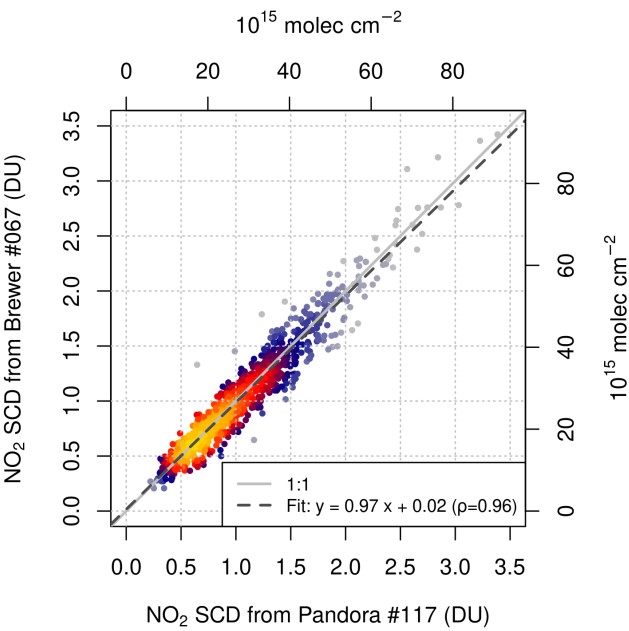

**Figure 7.** Comparison of the $NO_2$ slant column densities retrieved from Brewer #067 and Pandora #117 (considered as the reference) at APL (Rome). The Pearson's correlation coefficient between the two SCD series is $\rho = 0.96$. Notice that the range of the axes is wider than in Fig. 6. As explained in the main text, time matching between the two data sets is accomplished by considering retrievals within a 2-minute time difference (this also applies to the following figures).

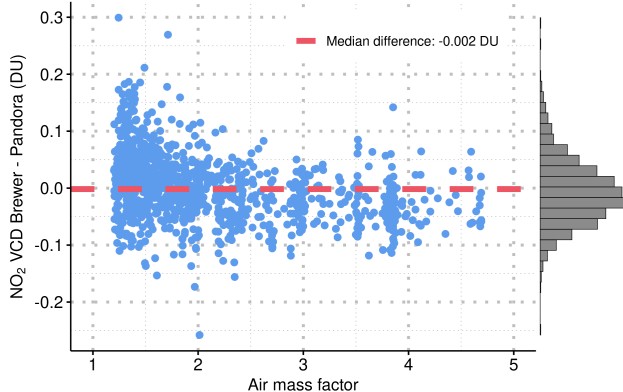

**Figure 8.** Difference between $NO_2$ VCD retrievals from Brewer #067 and Pandora #117 as a function of the air mass factor (AMF). The median of the differences (-0.002 DU or $-5.4 \times 10^{13}$ molec cm$^{-2}$) is plotted as a dashed line.

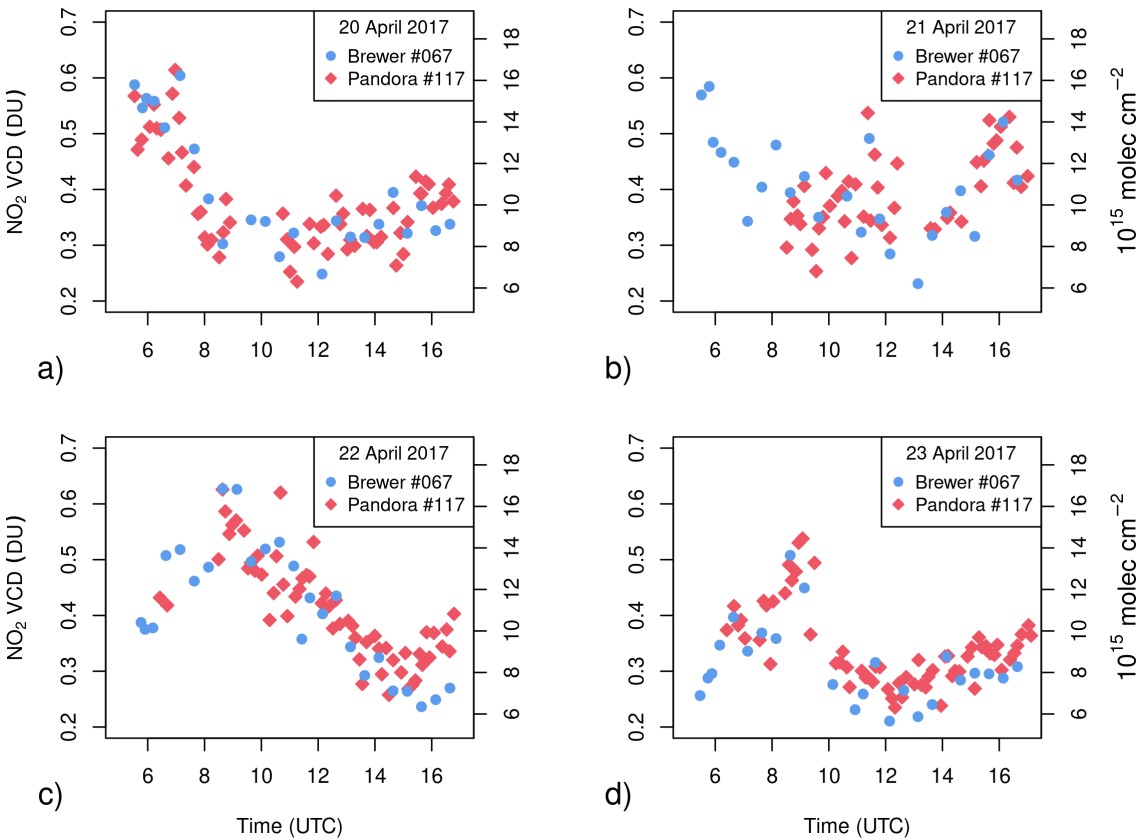

**Figure 9.** Independent retrievals from instantaneous measurements by Brewer #067 and Pandora #117 on four selected days (20–23 April 2017). Notice that some Pandora data are missing on 21 April before 8 UTC due to a power supply issue.

## 3.4 Comparison with Pandora

The second comparison proposed here is between Brewer #067 and Pandora #117 in Rome. The overlapping period of operation of both instruments extends from May 2016 to July 2017, and accounts for 185 fair weather days and more than 1000 simultaneous measurements (i.e., within a 2-minute time difference). Slant column densities (SCDs) are depicted in Fig. 7. Quite consistently with the results in Aosta–Saint-Christophe, the comparison shows a Pearson's correlation coefficient of 0.96 between the two sets (p-value $< 2.2{\times}10^{-16}$), a slope of 0.97 and an offset of 0.02 DU ($5.4{\times}10^{14}$ molec cm$^{-2}$). The differences between VCDs from Brewer and Pandora are represented in Fig. 8 as a function of the air mass, to show that their dependence on the air mass factor is negligible. The median VCD bias by Brewer #067 is $-0.002$ DU (-$5.4{\times}10^{13}$ molec cm$^{-2}$), clearly much lower than the combined uncertainty of both instruments, and 90 % of retrievals from Brewer #067 also fall within the nominal confidence levels of Pandora (0.1 DU or $2.7{\times}10^{15}$ molec cm$^{-2}$, 2-sigma estimate).

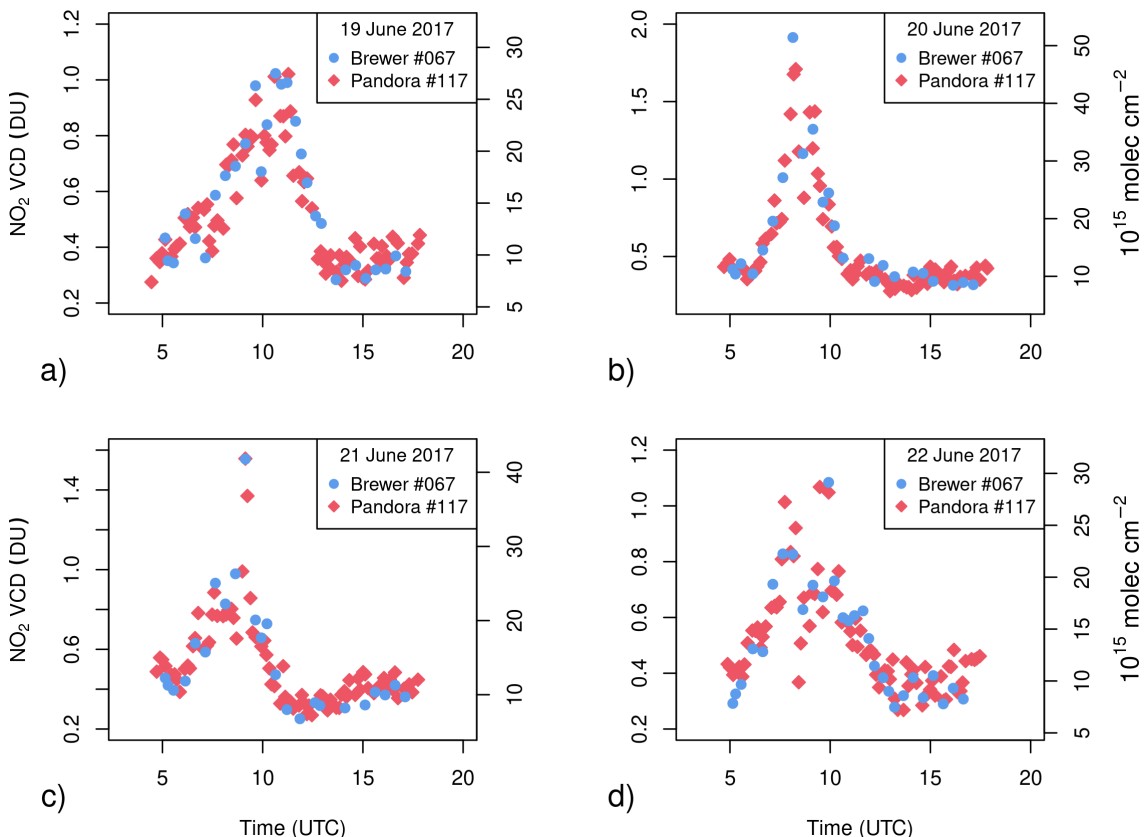

**Figure 10.** Independent retrievals from instantaneous measurements by Brewer #067 and Pandora #117 on four selected days (19-22 June 2017). Notice that the range of the vertical axis on 20 and 21 June is different from the other two days and that in subfigure c) the Brewer and Pandora measurements corresponding to the daily maximum (just before 10 UTC) perfectly overlap at about 1.5 DU and are not easily distinguishable.

To demonstrate the good agreement of the retrievals even in the short term (daily basis), two series of days in different seasons are chosen and the instantaneous data from Brewer #067 and Pandora #117 are displayed together (Figs. 9–10). These show that the $NO_2$ daily evolution is well captured by both instruments and that the absolute values of the two data sets are always well within the combined uncertainty of the measurements.

The possibility of comparing the two instruments also allows us to appreciate the better stability of BNALG2 compared to the
previous version of the algorithm, BNALG1. Figure 11 shows the difference between the retrievals from Brewer and Pandora on a portion of the data set, using both algorithms. In June 2017, the internal mercury lamp of Brewer #067, representing its wavelength reference, burned out and was replaced with a new one. This operation is sufficient to trigger an offset of 0.1 DU ($2.7 \times 10^{15}$ molec cm$^{-2}$) between the two instruments with BNALG1, which is very sensitive to small wavelength misalignments. The retrievals from BNALG2, instead, do not reflect such a large variation.

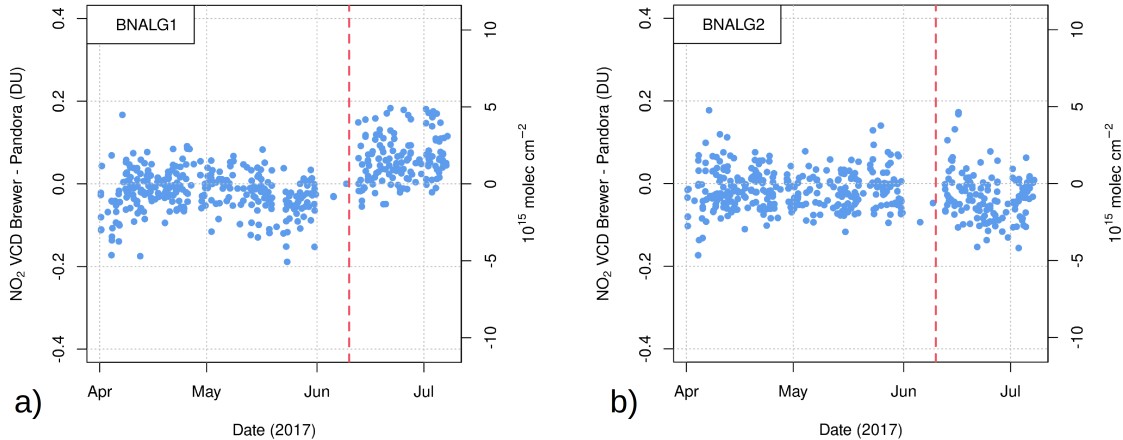

**Figure 11.** Difference between $NO_2$ retrievals from Brewer #067 – using BNALG1 (a) and BNALG2 (b) – and Pandora #117, on a portion of the data set. The vertical line shows the moment when the internal mercury lamp of Brewer #067, representing its wavelength reference, was replaced. This is sufficient to trigger a change with BNALG1, which is very sensitive to slight wavelength misalignments.

## 4 Discussion and conclusions

Although new generations of instruments have emerged in the last decades, and more specific, accurate and faster spectrometers have been made available to measure nitrogen dioxide in the atmosphere, the greatest asset from $NO_2$ data sets from Brewer spectrophotometers is the already existing network and the potential length of the recorded series. The data set from Rome represents an exemplary case, since the series covers – with impressive continuity and only few interruptions – the period from 1996 to the present, when important changes in tropospheric $NO_2$ emissions were expected to have occurred, especially in this European capital, in connection with technological upgrades (e.g., vehicle fleet and domestic heating sources), environmental control measures and other contingencies.

This justifies the effort made in this study to further improve the Brewer algorithm from BNALG1 to BNALG2. Experimental and modelling findings suggest that the data set obtained with the new method is less sensitive to instrumental interferences, such as wavelength misalignments and changes in the internal temperature, which also contributed to reduce the overall measurement uncertainty compared to RP2014. Additional innovations brought by BNALG2 are optimisation for polluted sites and use of the bootstrap estimation method to reconstruct the series of the extraterrestrial calibration coefficients and to track their temporal evolution. In this regard, a fortuitous example is offered by the change of the spectral radiometric sensitivity resulting from maintenance on the Brewer spherical mirror in July 2012, which is correctly captured by the bootstrap technique.

Although not abundant, and mostly concentrated in the last years of the data set, all available data collected by "reference" instrumentation operating alongside with Brewer #067, during intensive campaigns or much longer periods, were exploited to check the quality of our record. Comparisons with a Brewer (#066) calibrated using the Langley method at a pristine site and validated against NDACC instruments, and with a Pandora (#117) spectrometer show high and largely significant Pearson's

correlation coefficients ($>0.90$), slopes close to 1 and offsets well within the stated uncertainty. The latter was thoroughly calculated taking several instrumental, calibration and algorithmic factors into account, and assigned to each of the $>71000$ measurements. Since calibration provides the largest contribution to the overall uncertainty, one way of further reducing it is to establish new travelling reference instruments calibrated for $NO_2$ with the Langley technique, allowing harmonised retrievals within the network. A calibration protocol for $NO_2$ using a travelling standard MkIV Brewer is being developed by the authors using the current algorithm results. This will bring more consistency to the existing global Brewer $NO_2$ data and will make it easier to expand the usage of this algorithm in the future. Additionally, an ad hoc characterisation of MkIV Brewer spectrophotometers can be envisaged during international intercomparison campaigns (e.g., Redondas et al., 2018) to facilitate the implementation of our new algorithm and operational $NO_2$ retrievals on a network basis, such as in EUBREWNET (Rimmer et al., 2018).

This new data set can be used for calibration / validation exercises with satellite radiometers, comparison with models, and to calculate a climatology of the optical depth imputable to $NO_2$, which is useful, e.g., to improve the accuracy of aerosol optical depths from sun photometers. Further studies are needed to quantitatively determine the magnitude and the significance of the long-term trends that can be identified in the record in Rome, and their influencing factors. Since zenith sky measurements have also been recorded throughout the period of operation, they should be explored in future works, with the aim of developing and testing algorithms to partition the $NO_2$ column densities in the troposphere and in the stratosphere.

Thanks to the general formulation of the method and similar principles of operation of other MkIV spectrophotometers, the algorithm described here and applied to the record in Rome can be implemented at other stations of the international Brewer network, notably the ones with the longest historical data sets. As a confirmation of this, the new method has been successfully applied to the data set of Brewer #066 in Aosta–Saint-Christophe to assess the effect of the COVID-19 mobility restrictions on the $NO_2$ vertical column densities in 2020 (Diémoz et al., 2021).

# 5   Data availability

The data described in this manuscript can be freely accessed at https://doi.org/10.5281/zenodo.4715219 (Diémoz and Siani, 2021). The series is available as a Comma-Separated Value (CSV) file including vertical and slant column $NO_2$ densities, in different measurements units ($molec\,cm^{-2}$, $mol\,m^{-2}$ and DU), their respective expanded (k$=$ 2) uncertainties, and other ancillary information.

# Appendix A:  Detailed derivation of the Brewer equation

The extinction of the direct sun irradiance within the Earth atmosphere is described by the Bouguer-Lambert-Beer law (Bouguer, 1729), written here below in a general form. This will be solved in the following based on BNALG2 and by accounting for the most relevant species interacting with solar radiation in the MkIV Brewer measurement range:

$$\log I_i \quad = \quad \log I_{0_i} - \mu_{NO_2} X_{NO_2} \sigma_{NO_2 i}$$

$$-\mu_R \tau_{Ri}$$

$$-\mu_a \tau_{ai}$$

$$-\sum_j \mu_j \tau_{ji}$$

$$-D_{fi}$$

$$-\Lambda \tag{A1}$$

The spectral dependence of the equation is expressed by the index $i$, which refers to the considered wavelength ($\lambda_i$) among the ones measured by the Brewer (the total number of wavelengths extending up to 5 or 6, depending on the algorithm, cf. Tables 1 and 2). $I_i$ represents the count rates measured by the instrument at the Earth surface, proportional to the direct sun irradiance. These are assumed to have already undergone the preliminary data reduction (e.g., Kipp&Zonen, 2007; Siani et al., 2018), i.e. to have been corrected for the photomultiplier dark current, scaled for the integration time, and compensated for the deadtime from the photomultiplier and the counting system (electronics). $I_{0_i}$ is the count rate that would be measured if no extinction took place within the Earth atmosphere (Sect. 2.3.2). We denote with the subscripts $NO_2$, $R$, and $a$ the contributions by nitrogen dioxide, Rayleigh scattering, and aerosols, respectively. The other species interacting with solar radiation in the considered wavelength range (e.g., ozone, the oxygen dimer, water vapour, glyoxal, etc.) are marked with the subscript $j$. The airmass factors of a species, i.e. the enhancement of the light optical path, compared to the zenith, in the corresponding atmospheric layer at an effective height $h_{eff}$ is represented by $\mu$. The letter $X$ represents the VCD of a species, $\sigma_i$ its spectral cross section, and $\tau_i$ its optical depth (i.e., the product of VCD and cross section). In BNALG2, the cross sections are calculated as the convolution of high-resolution spectroscopic data sets with the instrumental slit function, by also accounting for the "I0-effect" (Aliwell et al., 2002), i.e. the fact that in actual measurements the spectra – and not their logarithm – are filtered by the slit function. Finally, the spectral attenuation of a density filter $f$ at wavelength $\lambda_i$ is denoted by $D_{fi}$ and the spectrally-invariant factors (such as the Earth-sun distance and, to a first approximation, clouds) by $\Lambda$.

We can rewrite Eq. (A1) once for each of the wavelengths collected in the course of a measurement. In this way, we obtain a system of equations, which can be solved for $X_{NO_2}$ using the addition method (linear combinations), i.e. by multiplying each equation by a coefficient $\gamma_i$ ("weighting"). The set of coefficients should be chosen in order to minimise the contribution of species other than $NO_2$. It should be noted that, since these weightings ultimately depend on the Brewer resolution and exact wavelengths, they can slightly change among instruments.

Unfortunately, in the above system there are fewer equations (measurement wavelengths) than the unknowns, therefore some of the interfering species will be cancelled out by the linear combination, others will be accounted for with a correction term,

and still others will be simply ignored based on their limited contribution. For example, in BNALG2 we use the following constraints to determine the weightings:

$$\begin{cases} \sum_i \gamma_i \equiv 0 \\ \sum_i \gamma_i \tau_{Ri} \equiv 0 \\ \sum_i \gamma_i \lambda_i^{-1} \equiv 0 \\ \sum_i \gamma_i \frac{\partial \log I}{\partial \lambda}\big|_{\lambda_i} \equiv 0 \end{cases} \tag{A2}$$

The first equation ensures that all spectrally flat factors ($\Lambda$) are correctly cancelled out in the linear combination. The second condition removes the effect by Rayleigh scattering. The third equation minimises the effect of Mie scattering by particles, assuming an aerosol Ångström exponent of 1 (i.e., $\tau_{ai} \equiv \beta \lambda_i^{-1}$). Notice that in the standard Brewer algorithm a linear dependence with $\lambda_i$ is used instead (i.e., $\sum_i \gamma_i \lambda_i \equiv 0$). The fourth condition minimises the algorithm sensitivity to small shifts in the Brewer wavelength scale (see main text for an in-depth discussion). This can be easily shown by linearising the logarithm of the measured irradiance around a wavelength $\lambda_i$:

$$\log I(\lambda_i + \Delta\lambda) \simeq \log I(\lambda_i) + \frac{\partial \log I}{\partial \lambda}\big|_{\lambda_i} \Delta\lambda \,, \tag{A3}$$

and by noting that, in this way, an extra term $\Delta\lambda \sum_i \gamma_i \frac{\partial \log I}{\partial \lambda}\big|_{\lambda_i}$ is introduced in the linear combination and must be minimised with a correct choice of the weightings. Here, the spectral derivative is calculated with the finite difference method by simulating the solar spectrum at ground with the libRadtran radiative transfer code (Emde et al., 2016) and shifting it by $\pm 2$ micrometer steps (corresponding to about 0.02 nm).

The constraints over the $\gamma_i$'s in Eq. (A2) form themselves a homogeneous system of equations. In order to avoid the trivial (null) solution, we must leave one degree of freedom available. Hence, we can only fix one additional constraint, which in BNALG2 we use to maximise the signal-to-noise ratio, as explained later on.

Setting aside, for a moment, the determination of the linear combination coefficients, we rewrite Eq. (A1) as follows:

$$\begin{aligned} \sum_i \gamma_i \log I_i \quad &+ \quad \sum_i \gamma_i \mu_{O_4} X_{O_4} \sigma_{O_4 i} \\ &+ \quad \sum_i \gamma_i D_{fi} \\ &= \quad \sum_i \gamma_i \log I_{0_i} - \sum_i \gamma_i \mu_{NO_2} X_{NO_2} \sigma_{NO_2 i} \\ &- \quad \sum_i \gamma_i \sum_u \mu_u \tau_{ui} \end{aligned} \tag{A4}$$

where the terms cancelled out or minimised by the linear combination are neglected, and the terms that can be interpreted as small corrections to the measured irradiance (i.e. oxygen dimer and filter transmittance) are moved to the left-hand side.

Indeed, once the Brewer has been characterised (Sect. 2.2.1) and the weightings are determined as described above, those terms can be easily calculated. Notably, in BNALG2 we estimate the contribution by the oxygen dimer based on surface pressure and the contribution by the density filter based on its spectral transmittance. The dependence of the Brewer spectral sensitivity on the instrument internal temperature (Sect. 2.3.1 and Fig. 1) can also be corrected at this stage. Finally, the species unaccounted for in BNALG2, e.g. ozone and water vapour ($u$ subscript), are included in the last term of Eq. (A4). Their effect is investigated in the main text.

After adjusting the linear combination of the measured irradiances for the correction terms (thus obtaining the corrected $\sum_i \gamma_i \log I_i^*$), the equation becomes:

$$
\begin{aligned}
\sum_i \gamma_i \log I_i^* \;=\;& \sum_i \gamma_i \log I_{0_i} - \mu_{NO_2} X_{NO_2} \sum_i \gamma_i \sigma_{NO_2 i} \\
& - \sum_i \gamma_i \sum_u \mu_u \tau_{ui}
\end{aligned}
\tag{A5}
$$

and the $NO_2$ VCD can be calculated as

$$
\begin{aligned}
X_{NO_2} \;=\;& \frac{\sum_i \gamma_i \log I_{0_i} - \sum_i \gamma_i \log I_i^*}{\mu_{NO_2} \sum_i \gamma_i \sigma_{NO_2 i}} \\
& - \epsilon_U
\end{aligned}
\tag{A6}
$$

where we have indicated with $\epsilon_U$ the effect of the unaccounted species. In other words, when neglecting this $\epsilon_U$ contribution, the error on the $NO_2$ VCD estimation will be

$$
\epsilon_U = \frac{\sum_i \gamma_i \sum_u \mu_u \tau_{ui}}{\mu_{NO_2} \sum_i \gamma_i \sigma_{NO_2 i}}
\tag{A7}
$$

Therefore, in order to minimise the influence of the unaccounted interfering species, we must maximise the $NO_2$ differential cross section, i.e. the term $\sum_i \gamma_i \sigma_{NO_2 i}$. This represents the fifth constraint in addition to the ones listed in Eq. (A2). The same condition also maximises the signal-to-noise ratio of the $NO_2$ retrieval. This is not demonstrated here for sake of brevity, but can be easily shown by propagating Poisson noise from the $I_i$ to $X_{NO_2}$ in Eq. (A6), as also described in Sect. 3.2.

*Author contributions.* HD and SC conceived this study, including comparison between the Brewer and reference instruments. HD re-evaluated the historical $NO_2$ data set of Brewer #067 and prepared the original draft of the paper. AMS is the responsible of the Brewer spectrophotometry activities in the frame of Sapienza Atmospheric Physics Laboratory and, together with GRC, has operated the Brewer for more than two decades and collected the raw data. VS has provided technical support with Brewers #066 and #067, and has calibrated and serviced Brewer #067 for ozone retrievals. AMI is the local PI for the Pandora spectrometer. AC is the PI of the Pandonia Global Network (PGN) and MT is responsible for the PGN algorithm development. MM performed instrument and $NO_2$ column calibrations for the Pandora. All authors have contributed to the preparation of the final version of the paper.

*Competing interests.* The authors declare that they have no conflict of interest.

*Acknowledgements.* The BAQUNIN project is funded by ESA (contract ID 4000111304/14/I-AM). The authors gratefully acknowledge ARPA Lazio for providing the $NO_2$ surface concentrations. The PGN (https://www.pandonia-global-network.org) is a bilateral project supported with funding from NASA and ESA.

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
