# Peer review of "Advanced $NO_2$ retrieval technique for the Brewer spectrophotometer applied to the 20-year record in Rome, Italy"

_Earth System Science Data, 2021_

## Author Comment (AC1)

**Response to Referee #1**

We thank the reviewer for taking the time to revise our manuscript and for his constructive comments. Our point-to-point reply is given hereafter (the text in italics represents a citation of the revised manuscript and the figure references follow the updated numbering).

**Referee's comment 1. The paper is mostly well written with minor language corrections needed (see attached document with corrections marked in green).**

Author's response 1. All language corrections (marked in green by the reviewer) were applied to the text.

**RC2. Figure 3. What are the corresponding Pandora measurements?**

AR2. Figure 3 was updated, it now includes the corresponding Pandora measurements. The new figure is reported here below.

[Figure]

**Figure 3:** VCDs retrieved using BNALG2 from Brewer #067: instantaneous measurements (grey dots), monthly and yearly averages (continuous lines). The monthly averages of the retrievals from Pandora #117 (Sect. 3.4), operating at APL since 2016, are also shown for comparison (dashed line).

**RC3. How is time matching accomplished in Figs. 8 and 9?**

AR3. We clarified that *"Time matching between the two data sets is accomplished by considering retrievals within a 2-minute time difference"* in the figure captions. This was already explained in the main text, which was not changed.

**RC4. The description of the equations used in the new algorithm is deficient. This needs to be fixed before publication. [...] As it is now, it is not possible to follow the derivation of equations 1 to 3 and more detail for the remaining equations 4 - 7.**

AR4. Section 2.3.1 (Algorithm) was rewritten and a new Appendix (Detailed derivation of the Brewer equation) was added at the end of the manuscript. Section 2.3.2 was expanded to provide more details on the calibration techniques. They are reported here below.

[revised manuscript text omitted]

**RC5. The comparison with Pandora 117 is quite good, However, even more interesting is a comparison of Brewer 067 with the daily files from Pandora 117.**

AR5. We thank the reviewer for spotting this. We have replicated his figures, as shown here below, and included them in the paper. Notice that Pandora measurements are missing on 21 April 2017 before 8 UTC due to power supply issues occurred during the night (as noted in the instrument logbook), not due to clouds.

[Figure]

**Figure 9:** Independent retrievals from instantaneous measurements by Brewer #067 and Pandora #117 on four selected days (20–23 April 2017). Notice that some Pandora data are missing on 21 April before 8 UTC due to a power supply issue.

[Figure]

**Figure 10:** Independent retrievals from instantaneous measurements by Brewer #067 and Pandora #117 on four selected days (19-22 June 2017). Notice that the range of the vertical axis on 20 and 21 June is different from the other two days and that in subfigure c) the Brewer and Pandora measurements corresponding to the daily maximum (just before 10 UTC) perfectly overlap at about 1.5 DU and are not easily distinguishable.

---

## Author Comment (AC2)

**Response to Referee #2**

We thank the reviewer for taking the time to revise our manuscript and for her constructive comments. Our point-to-point reply is given hereafter (the text in italics represents a citation of the revised manuscript).

**Referee's comment 1. I agree with the first reviewer – the equations need to be written out further to explain intermediate steps. If needed, this can be addressed in the Appendix.**

Author's response 1. Section 2.3.1 (Algorithm) was rewritten and a new Appendix (Detailed derivation of the Brewer equation) was added at the end of the manuscript. Section 2.3.2 was expanded to provide more details on the calibration techniques. They are reported here below.

[revised manuscript text omitted]